# Bandits with Knapsacks and Predictions

**Davide Drago**[1]        **Andrea Celli**[2]        **Marek Eliáš**[2]

[1] davide.drago@studbocconi.it, unaffiliated. During the research, Davide Drago was a student at Bocconi University.
[2] {andrea.celli2,marek.elias}@unibocconi.it Bocconi University, Milan, Italy

## Abstract

We study the Bandits with Knapsacks problem with the aim of designing a *learning-augmented* online learning algorithm upholding better regret guarantees than the state-of-the-art primal-dual algorithms with worst-case guarantees, under both stochastic and adversarial inputs. In the adversarial case, we obtain better competitive ratios when the input predictions are accurate, while also maintaining worst-case guarantees for imprecise predictions. We introduce two algorithms tailored for the full and bandit feedback settings, respectively. Both algorithms integrate a static prediction with a worst-case no-$\alpha$-regret algorithm. This yields an optimized competitive ratio of $(\pi + (1 - \pi)/\alpha)^{-1}$ in scenarios where the prediction is perfect, and a competitive ratio of $\alpha/(1-\pi)$ in the case of highly imprecise predictions, where $\pi \in (0, 1)$ is chosen by the learner and $\alpha$ is Slater's parameter. We complement this analysis by studying the stochastic setting under full feedback. We provide an algorithm which guarantees a pseudo-regret of $\widetilde{O}(\sqrt{T})$ with poor predictions, and 0 pseudo-regret with perfect predictions.

## 1 INTRODUCTION

In the *Bandits with Knapsacks* (BwK) framework a decision maker makes a sequence of $T$ decisions with the goal of maximizing their reward, while satisfying $m$ resource-consumption constraints [Badanidiyuru et al., 2018]. At each round $t$ up to $T$, the decision maker chooses an action $\xi_t$ and, subsequently, receive a reward $f_t(\xi_t) \in [0, 1]$ and incur in costs $c_t(\xi_t) \in [0, 1]^m$. The process stops at time horizon $T$, or when the cumulative costs $\sum_{t=1}^{T} c_{t,i}(\xi_t)$ exceed a given budget $B$ for at least one resource $i$.

The BwK model was extended in various directions such as studying the adversarial inputs [Immorlica et al., 2022], handling general types of objectives and constraints [Agrawal and Devanur, 2019], providing best-of-both-worlds guarantees [Castiglioni et al., 2022b], and studying contextual and combinatorial settings [Badanidiyuru et al., 2014, Agrawal et al., 2016, Sankararaman and Slivkins, 2018, Slivkins et al., 2023].

BwK has numerous applications such as dynamic pricing and online advertising [Besbes and Zeevi, 2009, Babaioff et al., 2012, Badanidiyuru et al., 2012, Wang et al., 2023, Feng et al., 2023]. In such scenarios, the online platforms overseeing these systems typically possess large amount of data that can be utilized for training machine learning models, enabling predictions of the future evolution of rewards and costs. Our objective is to enhance the performance of state-of-the-art primal-dual algorithms for BwK by incorporating such predictions. This brings us to the central research question of this paper: *can machine learning predictions enhance the performance of traditional primal-dual algorithms in the BwK framework?* To address this question, we present three novel algorithms: one designed for the stochastic setting, and the other two for the adversarial full and bandit case, respectively. When equipped with good predictions, our algorithms guarantee dramatic improvements in performance. Moreover, they recover known worst-case guarantees in case of imprecise predictions.

### 1.1 OUR RESULTS

We design two algorithms for adversarial case with full and bandit feedback respectively. In adversarial setting, the algorithms are evaluated based on their competitive ratio which is, roughly speaking, the ratio between the reward achieved by the best strategy in hindside and the reward achieved by the algorithm, see Section 2 for precise definition. Our algorithms use a prediction of the best fixed strategy in hindsight. Both algorithms achieve an optimal trade-off between consistency (performance with perfect prediction)

and robustness (performance with very bad prediction).

**Theorem 1.1.** *For adversarial setting with both bandit and full feedback, there is an algorithm which accepts a trade-off parameter $\pi \in (0,1)$ and, with high probability, achieves a competitive ratio $(\pi + (1-\pi)\rho)^{-1}$ with a perfect best fixed strategy prediction, and $1/(1-\pi)\rho$ with an arbitrarily bad prediction, where $\rho$ denotes the per-iteration budget.*

The trade-off parameter $\pi$ can be understood as a hyper-parameter chosen by the learner. Intuitively, our algorithm samples a part of the input to be handled according to the predicted strategy, and the rest of the input is handled using the worst-case $1/\rho$-competitive algorithm by Castiglioni et al. [2022b], which is based on the `LagrangeBwK` framework by Immorlica et al. [2019, 2022]. The main difficulty in the analysis lies in ensuring that our algorithms do not exceed their budget too early, missing out on possibly profitable items arriving towards the end of the input sequence. In case of bandit feedback, we need to handle difficulties in simulating the worst-case algorithm, since we cannot provide it with feedback in time steps handled according to the predicted strategy.

We show that the consistency-robustness trade-off in Theorem 1.1 is optimal up to a constant factor.

**Theorem 1.2.** *In the adversarial setting, no algorithm whose competitive ratio with perfect best fixed strategy prediction is at most $1/\pi$ can be $(1/2(1-\pi)\rho)$-competitive in the worst case.*

In order to complete the picture, we also propose an algorithm for stochastic setting with full feedback whose worst-case performance matches the optimal (without predictions) bound of Badanidiyuru et al. [2018]. To achieve this, we exploit the notion of *expected Lagrangian game* by Immorlica et al. [2022], and denote by $(\xi^*, \lambda^*)$ the Nash equilibrium strategies of such game. Then, given a prediction $(\xi^A, \lambda^A)$ on such strategies, we define the primal and dual prediction errors as $\eta^P = \text{TVD}(\xi^A, \xi^*)$ and $\eta^D = \|\lambda^A - \lambda^*\|_1$.

**Theorem 1.3.** *There is an algorithm for the stochastic setting such that, if provided with equilibrium predictions with error $\eta^P \leq \rho/\sqrt{T}$ and $\eta^D \leq 1/\sqrt{T}$, it achieves expected profit at least $\mathbb{E}[\text{OPT}] - \eta^P T$. Otherwise, its profit is at least $\text{REW} \geq \text{OPT} - \widetilde{O}(\sqrt{T})$.*

There is no need for a trade-off parameter in the preceding theorem compared to the adversarial setting: expected profit of our algorithm is optimal with a perfect predictions and smoothly deteriorates as the prediction error increases, eventually matching the bound by Badanidiyuru et al. [2018]. The dependency of the performance bound on the prediction error is called *smoothness*. Our algorithm tests the quality of the predictions while processing the input sequence, switching to the worst-case algorithm if their error seems high.

**Organization.** Sections 3 and 4 contain the description and analysis of our algorithms for the adversarial and the stochastic setting, respectively. Section 5 presents empirical evaluation of our algorithms. iof Theorem 1.2 can be found in Appendix 6.

## 1.2 RELATED WORK

We briefly survey works which are closely related to our paper. For additional references, readers are encouraged to consult Chapter 10 of [Slivkins et al., 2019], the survey by Mitzenmacher and Vassilvitskii [2022], and the website by Lindermayr and Megow [2023].

**Bandit with Knapsacks** The BwK problem, first presented by Badanidiyuru et al. [2013, 2018], serves as a foundational framework that melds the traditional bandit setting with resource-consumption constraints. Successive work by Agrawal and Devanur [2019] have extended this scope to handle more complex constraints and objectives. When considering Adversarial BwK, it is impossible to attain sublinear regret. Therefore, starting with the work by Immorlica et al. [2019, 2022], the focus has been providing no-$\alpha$-regret guarantees, where $\alpha$ is the *competitive ratio*. The initial guarantees of $\alpha = O(m \log T)$ by Immorlica et al. [2019] where later improved to a tight $O(\log m \log T)$ by Kesselheim and Singla [2020]. In the case of $B = \Omega(T)$, Castiglioni et al. [2022b] showed how to achieve a constant competitive ratio of $1/\rho$. This algorithm also guarantees $\widetilde{O}(\sqrt{T})$ regret in the stochastic setting. It employs a primal-dual template based on the `LagrangeBwK` framework by Immorlica et al. [2022] for optimizing the trade-off between maximizing rewards and efficient resource allocation. This will be the main building block for our framework. There were also recent works on this problem by Sivakumar et al. [2022] and Deb et al. [2024].

**Learning Augmented online algorithms** The framework of *learning-augmented* online algorithms was formally established by Lykouris and Vassilvtiskii [2018], a seminal work that later served as a basis for research on online algorithms with predictions by Mitzenmacher and Vassilvitskii [2020]. Applications of this framework are wide-ranging and include scheduling [Mitzenmacher, 2019, Lattanzi et al., 2020] and caching or paging algorithms [Lykouris and Vassilvtiskii, 2018, Rohatgi, 2020, Wei, 2020, Emek et al., 2021, Gupta et al., 2022, Antoniadis et al., 2023]. In addition, a general framework for integrating predictions into online primal-dual algorithms was introduced by Bamas et al. [2020]. Recently, Lyu and Cheung [2023] studied BwK in a setting with time-varying stochastic properties, with predictions of the next reward and cost provided at each time step, whereas our algorithms need only one prediction received in the beginning of the instance.

## 2 PRELIMINARIES

The learner has a finite set of $n$ actions $A$ and $m$ resources, and takes a sequence of decisions over $T$ rounds. We define the set of randomized strategies of the learner as $\Xi := \Delta(A)$. At each round $t$, an action $x_t$ is sampled according to $\xi_t \in \Xi$ and, subsequently, the environment yields a reward function $f_t : A \to [0, 1]$ and costs $c_t : A \to [0, 1]^m$. We denote a sequence of inputs $(f_t, c_t)_{t=1}^T$ as $\gamma_T$. At each round $t$, in the *full feedback* model the learner observes $f_t, c_t$, while in the *bandit feedback* model they only observe $f_t(x_t)$ and $c_t(x_t)$. Each resource $i \in [m]$ is endowed with a budget $B$.[1] The *per-iteration budget* is denotes as $\rho = B/T$. We will work under the same assumption of Castiglioni et al. [2022b] and Balseiro et al. [2022]: budgets are at least linear in time, i.e., $B = \Omega(T)$. The learning process stops at $T$ or as soon as one of the resources is fully depleted (i.e., the total consumption exceeds the budget). Following previous work, we assume the existence of a *void action* $\varnothing$ which guarantees 0 costs for each resource Badanidiyuru et al. [2018], Immorlica et al. [2022]. This guarantees the existence of a strictly feasible solution. A simple illustration of a void action can be found in the context of a repeated auction, where it consists of placing a bid of 0.

**Benchmarks**  In the adversarial setting, we employ as a baseline the total expected reward of the best *fixed* strategy in $\Xi$. Actions are drawn from the same fixed strategy until the time step $\tau$ in which the budget is fully depleted. The void action is played from $\tau + 1$ time until the end of the time horizon. Formally,

$$\mathrm{OPT} := \sup_{\xi \in \Xi} \mathbb{E}_{x \sim \xi} \left[ \sum_{t=1}^\tau f_t(x) \right],$$

where $\tau$ is also a random variable depending on the realized costs. For any randomized strategy $\xi$, $f_t(\xi) := \mathbb{E}_{x \sim \xi}[f_t(x)]$ ($c_t(\xi)$ is defined analogously as the expected cost with respect to $\xi$). The choice of this baseline is in line with previous work on the adversarial case by Immorlica et al. [2022], Castiglioni et al. [2022b]. We use competitive ratio as a performance measure. We say that an algorithm is $1/c$-competitive, if its reward is at least $c\mathrm{OPT} - o(T)$. Then, we also say that the algorithm achieves no-$1/c$-regret and call $1/c$ its competitive ratio.

When rewards and costs are stochastic the baseline is the best fixed randomized strategy that satisfies the constraints in expectation. This is the standard choice in stochastic BwK settings [Badanidiyuru et al., 2018, Castiglioni et al., 2022b]:

$$\mathrm{OPT} := \sup_{\xi \in \Xi} \mathbb{E}[f(\xi)] \quad \text{s.t.} \quad \mathbb{E}[c(\xi)] \le \rho,$$

[1]Considering $B_1 = \ldots = B_m = B$ is w.lo.g. as argued by Immorlica et al. [2022].

where expectations are with respect to randomness of the environment when generating rewards and costs. Performance of an algorithm with reward $\mathrm{REW}_{\mathrm{ALG}}$ is then measured using pseudo-regret, which is defined as $\mathrm{OPT} - \mathbb{E}[\mathrm{REW}_{\mathrm{ALG}}]$.

In order to simplify the notation, both benchmarks are denoted by $\mathrm{OPT}$, its meaning will be clear from context.

**Primal-dual template**  The standard solution to solve BwK problems in the adversarial case is resorting to a primal-dual algorithm based on the `LagrangeBwK` framework by Immorlica et al. [2022]. We will employ the version of Castiglioni et al. [2022b] described in Algorithm 1, since it provides guarantees both under adversarial and stochastic inputs. We denote such "worst-case" algorithm by $\mathrm{ALG}^{WC}$. The core idea of the `LagrangeBwK` is to define the Lagrange function

$$L : \mathcal{X} \times \mathcal{D} \ni (x, \lambda) \mapsto f_t(x) + \langle \lambda, \rho - c_t(x) \rangle,$$

with $\mathcal{D} = [0, 1/\rho]$, and to instantiate a primal and a dual regret minimizer, both having with no-regret guarantees in the adversarial case. The primal regret minimizer $R^P$ observes $L(\cdot)$ as its reward function at time $t$, while the dual regret minimizer $R^D$ observes rewards $-L(\cdot)$. The regret guarantees of the overall primal-dual framework are a byproduct of the interaction between these two regret minimizers. When needed, we will use the following two procedures: $\mathrm{ALG}^{WC}$.next_element() yields the new strategy $\xi$ which follows the worst-case algorithm, and $\mathrm{ALG}^{WC}$.observe_utility($f_t, c_t$) updates the internal state of $\mathrm{ALG}^{WC}$ using the environment's feedback.

---

**Algorithm 1:** $\mathrm{ALG}^{WC}$ [Castiglioni et al., 2022a]

**1 Input:** $\rho, T, R^P, R^D$;
**2 for** $t = 1, \ldots, T$ **do**
**3**  $\quad$ Get $x_t \sim \xi_t$ and $\lambda_t$ from $R^P$ and $R^D$, resp.;
**4**  $\quad$ **if** $\sum_{\tau=1}^t c_t(\xi_t) \ge \rho T - 1$ **then**
**5**  $\quad\quad$ $x_t \leftarrow \varnothing$;
**6**  $\quad$ Play $x_t$ and observe $f_t$ and $c_t$;
**7**  $\quad$ update $R^P$ and $R^D$ with rewards
    $\quad\quad r_t^P(\xi) := L(\xi, \lambda_t, f_t, c_t)$;
**8**  $\quad$ and $r_t^D(\lambda) := -L(\xi_t, \lambda, f_t, c_t)$, resp.;

---

Castiglioni et al. [2022b] proved the following two performance bounds for their algorithm.

**Proposition 2.1** (Thm. 6.1 of Castiglioni et al. [2022b]). *Let $\delta > 0$ and $\rho \in (0, 1)$ be constants. In the setting with constant budget per iteration $\rho$, the reward of Algorithm 1 in adversarial setting is at least*

$$\rho OPT - o(T)\sqrt{\log(1/\delta)}$$

*with probability at least $(1 - \delta)$*

**Proposition 2.2** (Thm. 7.1 of Castiglioni et al. [2022b]). *Denote* ALG *the profit of Algorithm 1 in stochastic setting with primal and dual regret minimizers whose guaranteed cumulative regret up to time $T$ is at most $\mathcal{E}_P(T)$ and $\mathcal{E}_D(T)$ respectively. For $\delta > 0$, with probability at least $1 - \delta$, we have* $\text{OPT}^{DP} - \text{ALG}$ *at most*

$$O(\rho^{-1}\sqrt{2T \log(mT/\delta)}) + \mathcal{E}_P(T) + \mathcal{E}_D(T),$$

*where the budget per iteration $\rho$ is a constant.*

To simplify the presentation, we present the results for $m = 1$ henceforth. The case of $m > 1$ can be addressed with minor modifications, as the guarantees of $\text{ALG}^{WC}$ remain applicable.

## 3   ADVERSARIAL SETTING

We propose two algorithms for the adversarial case: one for full feedback and one for bandit feedback. They are based on a simple idea of splitting the input into two subsequences by random sampling. One subsequence is then served by the predicted strategy, and the by Algorithm 1. In Section 6, we show that the our bounds are tight.

### 3.1   FULL-FEEDBACK ALGORITHM

Our algorithm splits the budget between the predicted policy $\xi^A$ and the worst-case $(1/\rho)$-competitive algorithm $\text{ALG}^{WC}$ based on the trade-off parameter $\pi$. The idea is to follow the predicted strategy $\xi^A$ in time steps sampled with probability $\pi$. In parallel, we simulate $\text{ALG}^{WC}$ as if it was serving the whole instance with the full budget $B$. We would like to follow the policy suggested by $\text{ALG}^{WC}$ with probability $(1 - \pi)$. However, there is one catch: costs and rewards may not be spread evenly over the time horizon in adversarial setting. Therefore, the predicted strategy and/or $\text{ALG}^{WC}$ may spend their part of the budget much earlier than if used exclusively on the whole instance with the full budget. This gap can be even linear in $T$. To prevent our algorithm from missing the items arriving in the last part of the input sequence, we sample the time steps served by $\xi^A$ and $\text{ALG}^{WC}$ with a slightly lower rate of roughly $\pi - 1/\sqrt{T}$ and $(1 - \pi - 1/\sqrt{T})$, respectively (see Algorithm 2). We describe our algorithm for the case with a single budget constraint. Generalization to $m$ budget constraints is straightforward and requires the choice of $\mu := \frac{2\sqrt{2\log(m/\delta)}}{\rho T^{1/2}}$. In Alg. 2, we write $x_t \sim \xi^I$ meaning that we can sample $x_t$ either from the prediction $\xi^A$ or from the strategy of $\text{ALG}^{WC}$ at time $t$, depending on the result of the sampling step.

We denote by $\tau_A$ (resp., $\tau_{WC}$) the time step when $B^A$ (resp., $B^{WC}$) is decreased below 1 (Line 8 of Alg. 2). Let $\tau_A^*$ and $\tau_{WC}^*$ denote the stopping times of the predicted strategy and of $\text{ALG}^{WC}$ respectively, when run on the whole input. The following lemma is crucial for our analysis.

---

**Algorithm 2:** Adversarial setting, full feedback

**1 Input:** $\xi^A$, $\pi$, $\delta$, $ALG^{WC}$, $B$;

**2** $B^A := \pi B$, $B^{WC} = (1 - \pi)B$, $\mu := \frac{2\sqrt{2\log(1/\delta)}}{\rho T^{1/2}}$;

**3 for** $t = 1, 2, \ldots, T$ **do**

**4**    $\xi_t^{WC} \leftarrow ALG^{WC}.\text{next\_element}()$;

**5**    Sample
    $I \sim p(\{A, WC, X\}) = (\pi - \mu, 1 - \pi - \mu, 2\mu)$;

**6**    **if** $I \in \{A, WC\}$ and $B_t^I \geq 1$ **then**

**7**       play $x_t \sim \xi^I$;

**8**       $B_t^I \leftarrow B_{t-1}^I - c_t(x_t)$;

**9**    **else**

**10**      play the void action;

**11**    $ALG^{WC}.\text{observe\_utility}(f_t, c_t)$;

---

**Lemma 3.1.** *Each inequality $\tau_A \geq \tau_A^*$ and $\tau_{WC} \geq \tau_{WC}^*$ holds with probability at least $(1 - \delta)$.*

*Proof.* The expected amount of budget used by policy $\xi^A$ until time $\tau_A^* - 1$ is

$$\sum_{t=1}^{\tau_A^*-1}(\pi - \mu)c_t^\top \xi^A \leq (\pi - \mu)(B - 1),$$

where $\pi - \mu$ is the probability that $\xi^A$ is being played and $c_t^\top \xi^A = \mathbb{E}[c_t(x_t)]$ is the expected cost in such case. Since $c_t(x_t) \in [0, 1]$ are independent random variables, Azuma–Hoeffding inequality implies the following: with probability at least $(1 - \delta)$, the real amount of budget used is larger than its expectation by at most $\sqrt{2\log(1/\delta)}\sqrt{T} \leq (\mu/2)\rho T < \mu(B - 1)$. Therefore, with the same probability, we have $B_{\tau_A^*-1}^A \geq 1$ and $\tau_A \geq \tau_A^*$.

Similarly, the expected amount of budget used by $ALG^{WC}$ until $\tau_{WC}^* - 1$ is

$$\sum_{t=1}^{\tau_{WC}^*-1}(1 - \lambda - \mu)c_t^\top \xi_t^{WC} \leq (1 - \pi - \mu)(B - 1).$$

Since random variables $c_t(x_t)$ form a martingale, by Azuma–Hoeffding inequality the real amount of budget used is larger than its expectation by at most $\sqrt{2\log(1/\delta)}\sqrt{T} \leq (\mu/2)\rho T < \mu(B - 1)$ with probability at least $(1 - \delta)$. Therefore, with the same probability, we have $B_{\tau_{WC}^*-1}^{WC} \geq 1$ and $\tau_{WC} \geq \tau_{WC}^*$. $\square$

Now, we are ready to prove the *consistency bound*, i.e., the bound which holds if the algorithm receives a perfect prediction.

**Lemma 3.2** (Consistency). *Given perfect prediction $\xi^A = \xi^*$, where $\xi^*$ denotes the best fixed strategy computed offline, Algorithm 2 achieves reward at least $(\pi + (1 - \pi)\rho)\text{OPT} - o(T)\sqrt{\log(1/\delta)}$ with probability at least $(1 - 4\delta)$.*

*Proof.* By Lemma 3.1, we have $\tau_A < \tau_A^*$ or $\tau_{WC} < \tau_{WC}^*$ with probability at most $2\delta$. Otherwise, we consider random

variables $X_t$ for $t = 1, \ldots, \tau_A^*$, such that $X_t = f_t(x_t) \in [0,1]$ if $\xi^A$ or $\xi_t^{WC}$ was played at time $t$ and zero otherwise. The reward of Algorithm 2 is then $\sum_{t=1}^T X_t$ whose expectation is at least

$$(\pi - \mu) \sum_{t=1}^{\tau_A^*} f_t^\top \xi^A + (1 - \pi - \mu) \sum_{t=1}^{\tau_{WC}^*} f_t^\top \xi_t^{WC}$$
$$\geq (\pi - \mu)\text{OPT} + (1 - \pi - \mu)(\rho\text{OPT} - o(T))$$
$$\geq (\pi + (1 - \pi)\rho)\text{OPT} - o(T),$$

The first inequality holds with probability $(1 - \delta)$ by Proposition 2.1. By Azuma–Hoeffding inequality $\sum_{t=1}^{\tau_A^*} X_t$ differs from its expectation by more than $\sqrt{T \log(1/\delta)}$ with probability less than $\delta$. Then, the reward of the algorithm is at least $(\pi + (1 - \pi)\rho)\text{OPT} - o(T)\sqrt{\log(1/\delta)}$ with probability at least $(1 - 4\delta)$. This concludes the proof. $\square$

The following lemma bounds the robustness of our algorithm, i.e., its performance with incorrect predictions.

**Lemma 3.3** (Robustness)**.** *Given an arbitrary prediction $\xi^A$, Algorithm 2 achieves expected reward at least $(1 - \pi)\rho\text{OPT} - o(T)\sqrt{\log(1/\delta)}$ with probability at least $(1 - 2\delta)$.*

*Proof.* By Lemma 3.5, we have $\tau_{WC} < \tau_{WC}^*$ with probability at most $\delta$. Otherwise, we consider random variables $X_t$ for $t = 1, \ldots, \tau_{WC}^*$, such that $X_t = f_t(x_t) \in [0,1]$ if $\xi_t^{WC}$ was played at time $t$ and zero otherwise. The reward of Algorithm 2 is then at least $\sum_{t=1}^{\tau_{WC}^*} X_t$ whose expectation is

$$(1 - \pi - \mu) \sum_{t=1}^{\tau_{WC}^*} f_t^\top \xi_t^{WC} = (1 - \pi - \mu)REW^{WC}$$
$$\geq (1 - \pi)\rho\text{OPT} - o(T).$$

By Azuma–Hoeffding inequality, $\sum_{t=1}^{\tau_{WC}^*} X_t$ differs from its expectation by more than $\sqrt{T \log(1/\delta)}$ with probability smaller than $\delta$.

Therefore, the reward of the algorithm is at least $(1 - \pi)\rho\text{OPT} - o(T)\sqrt{\log(1/\delta)}$ with probability at least $(1 - 2\delta)$. $\square$

The preceding bounds are both dependent on $\pi$. For $\pi = 1$, the algorithm is consistent but not robust, and vice versa for $\pi = 0$. Theorem 1.1 follows directly from the following statement:

**Theorem 3.4.** *Algorithm 2 run with parameters $\pi \in [0,1]$ and $\delta \in (0,1)$ satisfies the following statement with probability at least $(1 - 6\delta)$. If provided with a perfect prediction, its reward is at least $(\pi + (1 - \pi)\rho)\text{OPT} - o(T)$. Otherwise, its reward is at least $(1 - \pi)\rho\text{OPT} - o(T)$.*

*Proof.* Lemmas 3.2 and 3.3 fail with probability at most $4\delta$ and $2\delta$, respectively. Therefore, the statement holds with probability at least $(1 - 6\delta)$. $\square$

## 3.2 BANDIT FEEDBACK

In order to adapt Algorithm 2 to the bandit setting, we need to resolve the following issue: since we receive feedback only for actions which we have taken, we cannot simulate the worst-case algorithm $ALG^{WC}$ in parallel, as we did in the previous section. Instead, Algorithm 3 provides feedback to $ALG^{WC}$ only in time steps when its action was taken. In other time steps, it receives 0 reward and 0 cost. Our analysis then needs to deal with the fact that the performance guarantees of $ALG^{WC}$ do not apply to the input sequence as a whole but only to the sampled subsequence.

---

**Algorithm 3:** Adversarial setting, bandit feedback

**1 Input:** $\xi^A, \pi, \delta, ALG^{WC}, B$;

**2** $B^A := \pi B,\ B^{WC} = (1 - \pi)B,\ \mu := \frac{2\sqrt{2\log(1/\delta)}}{\rho T^{1/2}}$;

**3 for** $t = 1, 2, \ldots, T$ **do**

**4**      $\xi^{WC} := ALG^{WC}.\text{next\_element}()$;

**5**      Sample
       $I \sim p(\{A, WC, X\}) = (\pi - \mu, 1 - \pi - \mu, 2\mu)$;

**6**      **if** $I \in \{A, WC\}$ *and* $B_t^I \geq 1$ **then**

**7**          play $x_t \sim \xi^I$;

**8**          $B_t^I \leftarrow B_{t-1}^I - c_t(x_t)$;

**9**      **else**

**10**          play the void action;

**11**      **if** $I = WC$ **then**

**12**          $ALG^{WC}.\text{observe\_utility}(f_t(x_t), c_t(x_t))$;

**13**      **else**

**14**          $ALG^{WC}.\text{observe\_utility}(0,0)$;

---

Given input sequence $\gamma_t = (f_t, c_t)$ for $t = 1, \ldots, T$, we define $\tilde{\gamma}_t$ as $\gamma_t$ in rounds when $\xi^A$ was played and $(0,0)$ otherwise. Similarly, we define $\hat{\gamma}_t$ as $\gamma_t$ in rounds when $ALG^{WC}$ was played and $(0,0)$ otherwise. $ALG^{WC}$ is run on $\hat{\gamma}$ with budget $(1 - \pi)B$.

We denote $\tau(\xi, \gamma, B)$ the stopping time of strategy $\xi$ on input sequence $\gamma$ with budget $B$. In particular, we will be interested in $\tau_A = \tau(\xi^A, \tilde{\gamma}, \pi B)$, $\tau_A^* = \tau(\xi^A, \gamma, B)$, $\hat{\tau}^* = \tau(\xi^*, \hat{\gamma}, (1 - \pi)B)$, and $\tau^* = \tau(\xi^*, \gamma, B)$, where $\xi^*$ is the best fixed strategy computed offline.

The proof of the following lemma is similar to Lemma 3.1 and shows that the our algorithm does not run out of budget too early.

**Lemma 3.5.** *With probability at least $(1 - \delta)$, we have have $\tau_A \geq \tau_A^*$ and $\hat{\tau}^* \geq \tau^*$.*

*Proof.* The expected amount of budget used by policy $\xi^A$ until time $\tau_A^* - 1$ is

$$\sum_{t=1}^{\tau_A^*-1} (\pi - \mu) c_t^\top \xi^A \leq (\pi - \mu)(B - 1),$$

where $\pi - \mu$ is the probability that $\xi^A$ is being played and $c_t^\top \xi^A = \mathbb{E}[c_t(x_t)]$ is the expected cost in such case. Since $c_t(x_t) \in [0, 1]$ are independent random variables, the Azuma–Hoeffding inequality implies the following: with probability at least $(1 - \delta)$, the real amount of budget used is larger than its expectation by at most $\sqrt{2 \log(1/\delta)}\sqrt{T} \leq (\mu/2)\rho T < \mu(B - 1)$. Therefore, with the same probability, we have $B_{\tau_A^*-1}^A \geq 1$ and $\tau_A \geq \tau_A^*$.

The expected amount of budget used by policy $\xi^*$ until time $\tau^* - 1$ is

$$\sum_{t=1}^{\tau^*-1} (1 - \pi - \mu) c_t^\top \xi^* \leq (1 - \pi - \mu)(B - 1),$$

where $1 - \pi - \mu$ is the probability that $\hat{\gamma}_t \neq (0,0)$ and $c_t^\top \xi^* = \mathbb{E}[c_t(x_t)]$ is the expected cost in such case. Since $c_t(x_t) \in [0, 1]$ are independent random variables, Azuma–Hoeffding inequality implies the following: with probability at least $(1 - \delta)$, the real amount of budget used is larger than its expectation by at most $\sqrt{2 \log(1/\delta)}\sqrt{T} \leq (\mu/2)\rho T < \mu(B - 1)$. Therefore, with the same probability, we have $\hat{\tau}^* \geq \tau^*$. $\square$

In the following lemma, we compare the reward of $\text{ALG}^{WC}$ played on $\hat{\gamma}$ to the reward of the optimal solution $\text{OPT}_\gamma$ for the original instance $\gamma$.

**Lemma 3.6.** *With probability at least $(1 - \delta)$, the reward of $\text{ALG}^{WC}$ on input sequence $\hat{\gamma}$ with budget $(1 - \pi)B$ is at least*

$$\sum_{t=1}^T \hat{f}_t(x_t) \geq \rho(1 - \pi)\text{OPT}_\gamma - o(T)\log(1/\delta).$$

*Proof.* Consider a martingale $X_t = \hat{f}_t(x_t)$. We can express the reward of $\text{ALG}^{WC}$ on $\hat{\gamma}$ as $\sum_{t=1}^T X_t$. For each $\hat{\gamma}$, Proposition 2.1 implies that the following holds with probability at least $(1 - \delta)$:

$$\sum_{t=1}^T X_t \geq \rho \sum_{t=1}^\tau \hat{f}_t^\top \xi - o(T)\log(1/\delta) \quad \forall \xi \in \Xi.$$

Consider the expectation of $\sum_{t=1}^T X_t$ over the choice of $\hat{f}$. The preceding inequality implies

$$\mathbb{E}\left[\sum_{t=1}^T X_t\right] \geq \rho \sum_{t=1}^{\hat{\tau}^*} \mathbb{E}[\hat{f}_t]^\top \xi^* - o(T)$$

$$= \rho(1 - \pi - \mu) \sum_{t=1}^{\hat{\tau}^*} f_t^\top \xi^* - o(T)$$

$$\geq \rho(1 - \pi - \mu) \sum_{t=1}^{\tau^*} f_t^\top \xi^* - o(T)$$

$$= \rho(1 - \pi)\text{OPT} - o(T).$$

The first equality follows from $\mathbb{E}[\hat{f}_t] = (1 - \pi - \mu)f_t$. The second inequality holds with probability at least $(1 - \delta)$ because $\hat{\tau}^* \geq \tau^*$ by Lemma 3.5.

Since $X_t \in [0, 1]$ forms a martingale, Azuma–Hoeffding inequality implies that it diverges from its expectation by at most $\sqrt{T \log(1/\delta)}$ with probability $(1 - \delta)$. Therefore, with probability at least $(1 - 3\delta)$, we have $\sum_{t=1}^T X_t \geq \rho(1 - \pi)\text{OPT}_\gamma - o(T)\log(1/\delta)$. This concludes the proof. $\square$

The main consequence of the preceding lemma is our robustness bound.

**Lemma 3.7** (robustness). *Given an arbitrary prediction $\xi^A$, Algorithm 3 achieves expected reward at least $(1-\pi)\text{OPT} - o(T)\sqrt{\log(1/\delta)}$ with probability at least $(1 - 3\delta)$.*

*Proof.* This is a consequence of Lemma 3.6. The reward of Algorithm 3 is at least the reward of $ALG^{WC}$ on $\hat{\gamma}$, which is at least $\rho(1 - \pi)\text{OPT}_\gamma - o(T)\log(1/\delta)$ with probability at least $(1 - 3\delta)$. $\square$

The following lemma states the consistency bound of our algorithm.

**Lemma 3.8** (Consistency). *Given perfect prediction $\xi^A = \xi^*$, where $\xi^*$ denotes the best fixed strategy computed offline, Algorithm 3 achieves reward at least $(\pi + (1 - \pi)\rho)\text{OPT} - o(T)\sqrt{\log(1/\delta)}$ with probability at least $(1 - 4\delta)$.*

*Proof.* By Lemma 3.5, we have $\tau_A < \tau_A^*$ with probability at most $\delta$.

Otherwise, we consider random variables $X_t$ for $t = 1, \ldots, \tau_A^*$, such that $X_t = f_t(x_t) \in [0, 1]$ if $\xi^A$ was played at time $t$ and zero otherwise. The reward of Algorithm 2 then equal to the reward of the worst-case algorithm on $\hat{\gamma}$ plus $\sum_{t=1}^{\tau_A^*} X_t$, whose expectation is

$$(\pi - \mu) \sum_{t=1}^{\tau_A^*} f_t^\top \xi^A = (\pi - \mu)\text{OPT} \geq \pi\text{OPT} - o(T).$$

By Azuma–Hoeffding inequality, $\sum_{t=1}^{\tau_A^*} X_t$ differs from its expectation by more than $\sqrt{T \log(1/\delta)}$ with probability smaller than $\delta$.

Combining with Lemma 3.7 which holds with probability at least $(1 - 3\delta)$, the reward of the algorithm is at least $(1-\pi)\rho\text{OPT} - o(T)\sqrt{\log(1/\delta)} + \pi\text{OPT} - o(T)\sqrt{\log(1/\delta)}$ with probability at least $(1 - 4\delta)$. $\square$

Theorem 1.1 for the bandit feedback is implied by the following.

**Theorem 3.9.** *Algorithm 2 run with parameters $\pi \in [0, 1]$ and $\delta \in (0, 1)$ satisfies the following statement with*

*probability at least* $(1 - 7\delta)$. *If provided with a perfect prediction, its reward is at least* $(\pi + (1 - \pi)\rho)OPT - o(T \log(1/\delta))$. *Otherwise, its reward is at least* $(1 - \pi)\rho OPT - o(T \log(1/\delta))$.

*Proof.* Lemmas 3.8 and 3.7 fail with probabilities at most $4\delta$ and $3\delta$ respectively. Therefore, the statement holds with probability at least $(1 - 7\delta)$. □

# 4 STOCHASTIC SETTING

Our algorithm follows the framework of Castiglioni et al. [2022b] (see Section 2), with a custom optimizer which accepts an equilibrium prediction and suffers a smaller regret if this prediction is accurate enough. We call it "Check&switch optimizer", see Algorithm 4. It is designed to serve an arbitrary sequence $\ell_1, \ldots, \ell_T$ received online. The optimizer plays the predicted strategy during the first $\sqrt{T}$ rounds. Then, it evaluates its empirical regret $G(t)$ at each time step $t > \sqrt{T}$, to decide whether to switch to a classical worst-case optimizer or to keep playing the predicted strategy.

---
**Algorithm 4:** Check&switch optimizer
---
1 **Input:** $x^A$, $R^{WC}$;
2 **Initialize:** $\Delta := \sqrt{T}$, $h(t) := \frac{3}{\rho}\sqrt{2t \log 8T^2}$;
3 **for** $t = 1, \ldots, \Delta$ **do**
4     play strategy $x^A$;
5 **for** $t = \Delta + 1, \ldots, T$ **do**
6     $G(t) := \max_x \sum_{\tau=1}^{t-1}(\ell_\tau(x) - \ell_\tau(x^A))$;
7     **if** $G(t) \leq h(t)$ **then**
8         play predicted strategy $x_{t+1} = x^A$;
9     **else**
10         serve the rest of the input using a worst-case optimizer $R^{WC}$;
---

Algorithm 4 preserves the regret bounds of the optimizer $R^{WC}$ in the worst case:

**Observation 1.** *Let* $\mathcal{E}(T)$ *denote the regret of the worst-case algorithm* $R^{WC}$ *used by Algorithm 4. Then, the regret of Algorithm 4 is at most* $O(h(T) + \mathcal{E}(T))$.

*Proof.* Consider the last time step $s$ when the algorithm played $\xi^A$. The regret of the algorithm is at most $h(s-1) + 1 + \mathcal{E}(T - s)$. The total regret is therefore

$$h(s) + 1 + \mathcal{E}(T - s) \leq 1 + h(T) + \mathcal{E}(T). \quad □$$

We use the algorithm by Castiglioni et al. [2022b] (Algorithm 1) deploying Algorithm 4 with prediction $\xi^A$ as the primal algorithm, and with prediction $\lambda^A$ as the dual algorithm. This way, it will satisfy the following robustness bounds irrespective of the quality of the prediction $(\xi^A, \lambda^A)$.

---
**Algorithm 5:** Stochastic setting
---
Serve the input sequence using Algorithm 1, where:
$R^P$ is Algorithm 4 with prediction $\xi^A$
$R^D$ is Algorithm 4 with prediciton $\lambda^A$
---

**Lemma 4.1** (Robustness). *If the regret of the worst-case optimizers used by Algorithm 4 is* $\widetilde{O}(\sqrt{T})$, *then the pseudo-regret of Algorithm 5 is at most* $\widetilde{O}(\sqrt{T})$.

*Proof.* This follows directly from Observation 1. Indeed, $h(T) = \widetilde{O}(\sqrt{T})$ and Algorithm 4 using a worst case optimizer with regret $\tilde{O}(\sqrt{T})$ have regret at most $\widetilde{O}(\sqrt{T})$. The lemma then follows from Proposition 2.2. □

## 4.1 SMOOTHNESS AND CONSISTENCY

We recall that the Lagrangian at time $t$ is defined as $L(\xi, \lambda, f_t, c_t) = f_t^\top \xi + \lambda_t(\rho - c_t^\top \xi)$. Moreover, let $\bar{f} := \mathbb{E}[f_t]$ and $\bar{c} := \mathbb{E}[c_t]$. The following technical lemma will be useful to estimate the probability of Algorithm 5 using the predicted strategy during the whole request sequence.

**Lemma 4.2.** *Consider arbitrary* $\xi \in \Xi, \lambda \in \mathcal{D}, \tau \in \mathbb{N}$. *We have*

$$|\sum_{t=1}^{\tau} L(\xi, \lambda, f_t, c_t) - \sum_{t=1}^{\tau} L(\xi, \lambda, \bar{f}, \bar{c})| \leq h(\tau) \quad (4.1)$$

*with probability at least* $1 - 1/4T^2$.

*Proof.* We define $X_\tau = \frac{\rho}{3} \sum_{t=1}^{\tau} (L(\xi, \lambda, f_t, c_t) - L(\xi, \lambda, \bar{f}, \bar{c}))$. Note that $f_t$ and $c_t$ are i.i.d. samples for each $t$. Therefore, we have

$$\mathbb{E}[X_\tau \mid X_1, \ldots, X_{\tau-1}]$$
$$= X_{\tau-1} + \frac{\rho}{3}\mathbb{E}[L(\xi, \lambda, f_\tau, c_\tau) - L(\xi, \lambda, \bar{f}, \bar{c})]$$
$$= X_{\tau-1},$$

i.e., $X_\tau$ is a martingale. The last equality holds because $\mathbb{E}[f_\tau] = \bar{f}$ and $\mathbb{E}[c_\tau] = \bar{c}$ and $L$ is linear in both $f$ and $c$, implying $\mathbb{E}[L(\xi, \lambda, f_\tau, c_\tau) - L(\xi, \lambda, \bar{f}, \bar{c})] = 0$. Moreover, $|X_\tau - X_{\tau-1}| = \frac{\rho}{3}|L(\xi, \lambda, f_\tau, c_\tau) - L(\xi, \lambda, \bar{f}, \bar{c})| \leq 1$, because $\lambda \in [0, 1/\rho]$ and therefore $L(\xi, \lambda, f, c) \in [0, 3/\rho]$ for any $f$ and $c$.

By Azuma–Hoeffding inequality, we have $\mathbb{P}(X_\tau > \alpha\sqrt{\tau}) < \exp(-\alpha^2/2)$. Substituting $X_\tau$ and choosing $\alpha := \frac{\rho}{3}h(\tau)/\sqrt{\tau} = \sqrt{2\log(8T^2)}$, we can express the probability of (4.1) being false as

$$\mathbb{P}\left(\frac{\rho}{3}|\sum_{t=1}^{\tau} L(\xi, \lambda, f_t, c_t) - \sum_{t=1}^{\tau} L(\xi, \lambda, \bar{f}, \bar{c})| \leq \frac{\rho}{3}h(\tau)\right)$$
$$< 2\exp(-\log(8)) = 4\delta. \quad □$$

When playing a predicted strategy, the difference of our rewards and costs from the strategy $(\xi^*, \lambda^*)$ is proportional to prediction error:

**Lemma 4.3.** *Let $x_1, \ldots, x_T$ be a sequence of vectors with entries from interval $[0, \alpha]$. For each $\tau = 1, \ldots, T$, we have $|\sum_{t=1}^{\tau} x_t^\top \xi^A - \sum_{t=1}^{\tau} x_t^\top \xi^*| \leq \tau \alpha \eta^P$ and $|\sum_{t=1}^{\tau} x_t^\top \lambda^A - \sum_{t=1}^{\tau} x_t^\top \lambda^*| \leq \tau \alpha \eta^D$.*

*Proof.* For each $t$, we have $|x_t^\top (\xi^A - \xi^*)| \leq \alpha TVD(\xi^A, \xi^*) = \alpha \eta^P$ and $x_t^\top (\lambda^A - \lambda^*) \leq \alpha \|\lambda^A - \lambda^*\|_1 = \alpha \eta^D$. $\square$

The following key lemma states that if both $\eta^P$ and $\eta^D$ are small, our algorithm will never switch to the worst-case algorithm.

**Lemma 4.4.** *If $\eta^P \leq \rho/\sqrt{T}$ and $\eta^D \leq 1/\sqrt{T}$, then Algorithm 5 plays strategy $(\xi^A, \lambda^A)$ at each step with probability at least $1 - 1/T$.*

*Proof.* For any time $\tau = 1, \ldots, T$, we show that with probability at least $(1 - 1/T^2)$, both conditions $G^P(\tau) = \max_\xi \sum_{t=1}^{\tau} (\ell_t^P(\xi) - \ell_t^P(\xi^A)) \leq 3h(\tau, \delta/4T)$ and $G^D(\tau) = \max_\lambda \sum_{t=1}^{\tau} (\ell_t^D(\lambda) - \ell_t^D(\lambda^A)) \leq 3h(\tau, \delta/4T)$ checked by Algorithm 4 hold. By union bound, the predicted solutions are therefore used during the whole runtime with probability at least $1 - 1/T$.

First, we check the primal condition. For any possible maximizer $\xi^E$ of $G^P$, we have

$$\sum_{t=1}^{\tau} L(\xi^E, \lambda^A, f_t, c_t) \leq \sum_{t=1}^{\tau} L(\xi^E, \lambda^*, f_t, c_t) + \tau \eta^D$$

$$\leq \sum_{t=1}^{\tau} L(\xi^E, \lambda^*, \bar{f}, \bar{c}) + h(\tau) + \tau \eta^D$$

$$\leq \sum_{t=1}^{\tau} L(\xi^*, \lambda^*, \bar{f}, \bar{c}) + h(\tau) + \tau \eta^D$$

$$\leq \sum_{t=1}^{\tau} L(\xi^*, \lambda^*, f_t, c_t) + 2h(\tau) + \tau \eta^D$$

$$\leq \sum_{t=1}^{\tau} L(\xi^A, \lambda^A, f_t, c_t) + 2h(\tau) + 2\tau \eta^D$$

$$+ \tau(1 + 1/\rho)\eta^P \leq 3h(\tau).$$

Each inequality 2 and 4 holds with probability at least $1 - 1/4T^2$ by Lemma 4.2. Inequalities 1 and 5 follow from Lemma 4.3 and Inequality 3 holds because $(\xi^*, \lambda^*)$ is the equilibrium strategy and $\xi^*$ is therefore the best response to $\lambda^*$ in $L(\xi, \lambda^*, \bar{f}, \bar{c})$. The last inequality then follows from the assumption about the size of $\eta^P$ and $\eta^D$. Since $\ell_t^P(\xi) = L(\xi, \lambda^A, f_t, c_t)$, we have the desired bound.

Similarly, we have

$$-\sum_{t=1}^{\tau} L(\xi^A, \lambda^E, f_t, c_t) \leq -\sum_{t=1}^{\tau} L(\xi^*, \lambda^E, f_t, c_t) + \tau \eta^P$$

$$\leq -\sum_{t=1}^{\tau} L(\xi^*, \lambda^E, \bar{f}, \bar{c}) + h(\tau) + \tau \eta^P$$

$$\leq -\sum_{t=1}^{\tau} L(\xi^*, \lambda^*, \bar{f}, \bar{c}) + h(\tau) + \tau \eta^P$$

$$\leq -\sum_{t=1}^{\tau} L(\xi^*, \lambda^*, f_t, c_t) + 2h(\tau) + \tau \eta^P$$

$$\leq -\sum_{t=1}^{\tau} L(\xi^A, \lambda^A, f_t, c_t) + 2h(\tau) + \tau \eta^D$$

$$+ \tau(2 + 1/\rho)\eta^P \leq 3h(\tau),$$

where the second and the fourth inequality each hold with probability $1 - 4T^2$. To get the statement of the lemma, it is enough to use the union bound. $\square$

In order to prove the smoothness bound, we need to consider the following. If the algorithm follows $\xi^A$ which slightly differs from the equilibrium strategy $\xi^*$, it may run out of budget earlier than $\xi^*$: at that moment, an algorithm using $\xi^*$ at each time step is left with a fraction of its budget and can use it to acquire further reward. To bound this reward, we use the following lemma.

**Lemma 4.5.** *Consider $\tau \in \{1, \ldots, T\}$ and an input subsequence $\bar{\gamma} = \gamma_\tau, \ldots, \gamma_T$. For $\alpha \in [0, 1]$, the expected reward, over the random choice of the original sequence $\gamma$, of the strategy $\xi^*$ on $\bar{\gamma}$ with budget $\alpha B$ is at most $O((\alpha/\bar{c}(\xi^*))T \log T)$.*

*Proof.* If $\alpha B < 1$, the reward of strategy $\xi^*$ is 0 and the lemma follows. Therefore, it is enough to consider $\alpha B \geq 1$.

We choose $k = \bar{c}(\xi^*)^{-2}(4 \log T) \alpha B$ and show that with probability $(1 - 1/T)$, the strategy $\xi^*$ exceeds its budget before $k$ time steps. For any $\tau \in \{1, \ldots, T - k\}$, we have

$$\mathbb{E}\left[\sum_{t=\tau}^{\tau+k} c_t(\xi^*)\right] = k\bar{c}(\xi^*) \geq \frac{4 \log T}{\bar{c}(\xi^*)} \alpha B.$$

With probability $(1 - 1/T)$, $\sum_{t=\tau}^{\tau+k} c_t(\xi^*)$ is smaller than its expectation by at most $\sqrt{2k \log T}$, i.e., we have

$$\sum_{t=\tau}^{\tau+k} c_t(\xi^*) \geq \frac{4 \log T}{\bar{c}(\xi^*)} \alpha B - \frac{\sqrt{8} \log T}{\bar{c}(\xi^*)} \sqrt{\alpha B} > \alpha B.$$

Otherwise, the reward of $\xi^*$ is clearly at most $T$.

Therefore, the expected reward of $\xi^*$ is at most

$$\left(1 - \frac{1}{T}\right) k\bar{f}(\xi^*) + \frac{1}{T}T \leq k + 1 \leq O\left(\frac{\alpha}{\bar{c}(\xi^*)} T \log T\right).$$
$\square$

**Lemma 4.6.** *Given a prediction with primal error $\eta^P \leq \rho/\sqrt{T}$ and $\eta^D \leq 1\sqrt{T}$, the pseudo-regret of our algorithm is at most $\tilde{O}(\eta^P T/\rho^2)$.*

*Proof.* We analyze the case when our algorithm plays $\xi^A$ at each time step. Otherwise, (this happens with probability at most $1/T$) it is enough to bound its pseudo-regret by $T$.

Let $\tau^A$ and $\tau^*$ denote the stopping times of the strategies $\xi^A$ and $\xi^*$ respectively. We claim that

$$\mathbb{E}[\sum_{t=1}^{\tau^*} f_t^\top \xi^* - \sum_{t=1}^{\tau_A} f_t^\top \xi^A]$$
$$\leq \mathbb{E}[\sum_{t=1}^{\tau_A} f_t^\top (\xi^* - \xi^A)] + O(\frac{\eta^P}{\rho^2} T \log T),$$

where the expectation in the right-hand side can be bounded by $T\eta^P$ (Lemma 4.3). To show that the inequality holds, we proceed as follows.

Lemma 4.3 implies that $\sum_{t=1}^{\tau_A} c_t^\top \xi^* \geq \sum_{t=1}^{\tau_A} c_t^\top \xi^A - \tau^A \eta^P$, i.e., the leftover budget of the strategy $\xi^*$ at time $\tau_A$ is at most $\eta^P T = \frac{\eta^P}{\rho} B$. Therefore, by Lemma 4.5, the expected reward of strategy $\xi^*$ after $\tau_A$ is bounded by $\tilde{O}(\eta^P T/\rho^2)$ whenever $\bar{c}(\xi^*) \geq \rho/2$. If this is not the case, $\tau^A = \tau^* = T$ with probability at least $(1 - 1/T)$ and we have $\mathbb{E}[\sum_{t=1}^{\tau^*} f_t^\top \xi^* - \sum_{t=1}^{\tau_A} f_t^\top \xi^A] \leq \eta^P T$ by Lemma 4.3. $\qquad\square$

**Consistency** If $\xi^A = \xi^*$ and $\lambda^A = \lambda^*$, Algorithm 5 plays $\xi^A$ at every time step by Lemma 4.4 with probability $(1 - 1/T)$ and its pseudo-regret with respect to $\xi^*$ is 0.

## 5 EMPIRICAL RESULTS

We have implemented[2] our algorithms for adversarial setting (Algorithms 2 and 3) and the worst-case algorithm of Castiglioni et al. [2022b] which we denote Primal-Dual. We have compared their performance both in full feedback and bandit setting in several experiments on stochastic instances generated as follows: There are 5 actions, $T = 50\,000$ time steps, $\rho = 0.1$. Action 5 is the void action with reward 0 and cost 0 at each time step. The rewards and costs of actions $1, \ldots, 4$ at each time $t = 1, \ldots, T$ are drawn independently from a log-normal distribution with parameter $\sigma = 0.5$.[3] The $\mu$ parameter of the rewards of actions $1, \ldots, 4$ is $0.9, 1.2, 0.1$, and $0.4$ respectively. The $\mu$ parameter of the costs of actions $1, \ldots, 4$ is $0.8, 0.6, 1.2$, and $0.5$ respectively. Since the problem requires bounded rewards and costs, we

---

[2] https://github.com/davidedrago0007/
AdversarialBanditwithKnapsacksandPredictions

[3] The choice of log-normal distributions is motivated by applications in Internet advertising, where they are often used to model bidder's valuations (see, e.g., [Balseiro et al., 2020]).

| Algorithm | Reward | Cost |
|---|---|---|
| Primal-Dual (full feedback) | 6111.29 | 4256.78 |
| Algorithm 2 (full feedback) | 8060.92 | 4897.84 |
| Primal-Dual (bandit feedback) | 4537.85 | 3825.01 |
| Algorithm 3 (bandit feedback) | 8099.34 | 4983.35 |

**Figure 1:** Performance with perfect prediction

replace each reward and cost exceeding 5 by 5. Ultimately, we normalize the data in order to keep the values between 0 and 1 to remain consistent with our setting. In each experiment, the instance was generated in advance and then provided to all algorithms in an online manner.

**Results with perfect predictions** With perfect predictions, our algorithm outperforms Primal-Dual by Castiglioni et al. [2022b] both in full feedback and bandit setting. Figure 1 contains comparison of rewards achieved by Primal-Dual and our Algorithms 2 and 3. With every experiment, our algorithms received prediction $\xi^A = \xi^*$, where $\xi^*$ is the best fixed strategy for the generated input instance computed offline. Our algorithms were both implemented with $\pi = 0.9$ and $\mu = 0$. This choice of $\mu$ seems to be suitable for stochastic inputs, although our analysis requires a higher $\mu$ in order to avoid overspending in highly adversarial cases. Appendix A contains results with different choices of $\mu$ and $\pi$. In particular, our algorithms can achieve improvement over Primal-Dual already with $\pi = 0.5$.

Values in Figure 1 are averaged over 10 independent experiments and the standard deviation was $41.56$ and $21.70$ in the full feedback setting for Primal-Dual and Algorithm 2 respectively. In the bandit setting, Primal-Dual presented a concerning standard deviation of $2843.01$ signaling instability of the algorithm, while Algorithm 3 showed a standard deviation of $58.43$.

One may notice that Algorithm 3 for the bandit setting achieves a slightly better reward than Algorithm 2. This seems to be caused by feedback $(0, 0)$ provided to the worst-case algorithm in Line 14 of Algorithm 3 steering down the dual variables. This allows for higher spending, which gives the algorithm an advantage in our experiments.

**Results with noisy predictions** We ran 10 independent experiments. In each experiment, we first computed the best fixed solution and added to it an independent noise coming from a gaussian distribution with $\sigma \in [0, 0.1]$. The prediction provided to the algorithm was a normalization of the resulting vector. With $\sigma = 0.1$, the average prediction error was $0.19$.

Figures 2 and 3 show that of our algorithms improve over Primal-Dual for small $\sigma$. The figures contain average rewards of 10 experiments, with the lowest standard deviation of $21.7$ in the case of $\sigma = 0.0$ increasing up to $330.02$ for

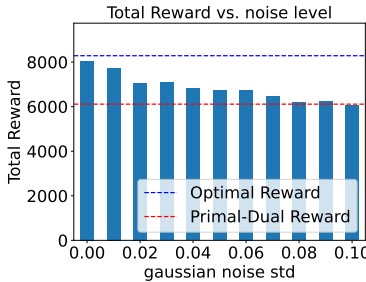

**Figure 2:** Full feedback for different noise levels with $T = 50000, B = 5000, \pi = 0.9$

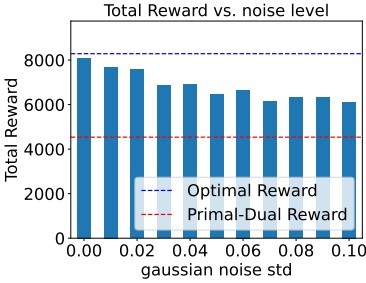

**Figure 3:** Bandit feedback for different noise levels with $T = 50000, B = 5000, \pi = 0.9$

$\sigma = 0.03$; the highest standard deviation is registered in the case of $\sigma = 0.7$, with a value of $851.68$. In the bandit setting, Algorithm 3 outperforms Primal-Dual with any level of noise shown in the graph. This is due to the bandit algorithm not being able to spend the full budget, as already shown in table 1. With $\sigma$ much larger, the performance of our algorithm was at most 30 percent lower than Primal-Dual in the full feedback and almost comparable in the bandit setting, see Appendix A.

**Additional experiments**    Appendix A contains additional experimental results which detail budget spending and cumulative rewards by implemented algorithms and results achieved by our algorithm with different parameters.

# 6   LOWER BOUND

**Theorem 6.1.** *Consider arbitrary constants $\pi, \epsilon \in (0, 1/4)$. In equilibrium-prediction setup, no $1/\pi$-consistent algorithm can be $1/(1 + \epsilon)(1 - \pi)$-robust.*

*Proof.* Consider two input instances with $\rho = 1/2$ and two non-void actions. The algorithm receives prediction $\xi^A = (1, 0)$, which is perfect for the first instance and incorrect for the second one.

For $t = 1, \dots, T/2$, we have $f_t(1) = \epsilon$, $c_t(1) = 1$ and $f_t(2) = c_t(2) = 0$ in both instances. For $t = T/2 + 1, \dots, T$, the two instances differ: we have $f_t(1) =$

$c_t(1) = 0$ and $f_t(2) = c_t(2) = 0$ in the first instance and $f_t(1) = c_t(1) = 0$ and $f_t(2) = c_t(2) = 1$ in the second instance.

Prediction $\xi^A$ is prefect for the first instance and any $1/\pi$-consistent algorithm has to take the first action at least $\pi T/2$-times until time $T/2$. Its remaining budget at $T/2$ is therefore at most $(1 - \pi)T/2$. Its total reward on the second instance can be at most $\epsilon \pi T/2 + (1 - \pi)T/2 < (1 + \epsilon)(1 - \pi)T/2$, where $T/2$ is the reward of the best fixed strategy $(0, 1)$. $\square$

# 7   CONCLUSIONS

In this paper, we provide ML-augmented algorithms for the adversarial setting of Bandits with Knapsack achieving the optimal trade-off between consistency and robustness in both full-feedback and bandit-feedback setting.

For stochastic setting, we propose a consistent, smooth, and robust algorithm which works in full-feedback setting. Optimal smoothness bounds for full-feedback setting and extension to bandit-feedback setting remain interesting open questions.

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

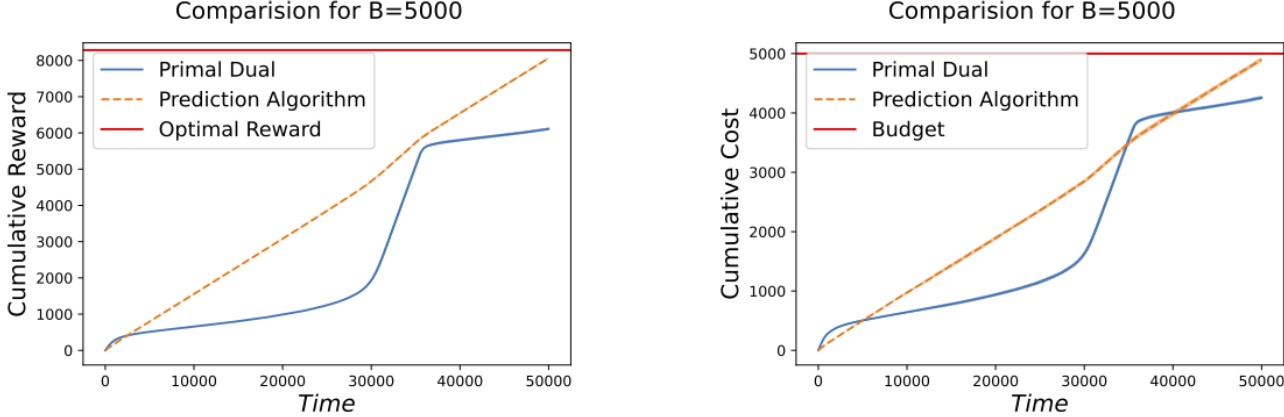

**Figure 4:** Experiment with $T = 50000, B = 5000, \pi = 0.9$. **Right:** FF Reward. **Left:** FF Cost.

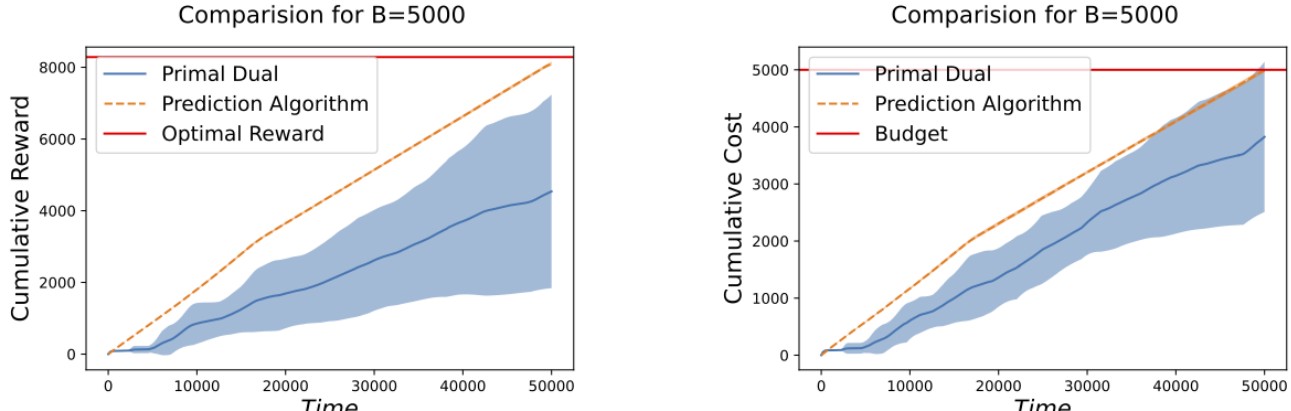

**Figure 5:** Experiments with $T = 50000, B = 5000, \pi = 0.9$. **Right:** Bandit setting, rewards. **Left:** Bandit setting, cost.

## A   ADDITIONAL EXPERIMENTAL RESULTS

### A.1   REWARD ACROSS HORIZON

In this section we present more in depth results linked to the experiments in Figure 1. For each algorithm in the table we show both the rewards and costs over time. In Figure 4 we can visualize the comparison between the two algorithms in the full feedback setting. The choice of $\pi = 0.9$, combined with the stochasticity of the data results in a line increasing linearly in both rewards and costs. The primal-dual algorithm, in blue, has an interesting behavior. First, it explores by spending a low amount of budget, then - around iteration 30000 - it starts heavily exploiting the acquired knowledge. However, the amount of budget spent increases fast, making the dual term prevalent, and not allowing to fully spend the remaining part of the budget. This last interaction causes the primal-dual algorithm to perform worse than Algorithm 2. In the bandit setting (Figure 5), Algorithm 3 presents a very similar behavior to the previous results. On the other hand, the primal-dual algorithm in bandit setting is not able to consistently provide good results, and presents a large variance throughout the entire time horizon. In Figure 6, we experimented on the case where $\pi = 0.5$. In such case we clearly expect a decrease in performance and an increase in uncertainty, especially in the case of the bandit setting. Both behaviors are shown in the graphs. In Figure 6, the full feedback setting is presented. Here, Algorithm 2 is still able to achieve a better performance than the primal-dual one, however with a lower total reward with respect to the $\pi = 0.9$ case.

In Figure 7 we observe how the decrease in the use of the prediction decreases the overall performance of Algorithm 3. Moreover, there is a significant increase in variance due to the variability which can be attributed to the primal-dual algorithm. Overall, the performance is still higher compared to the primal-dual benchmark.

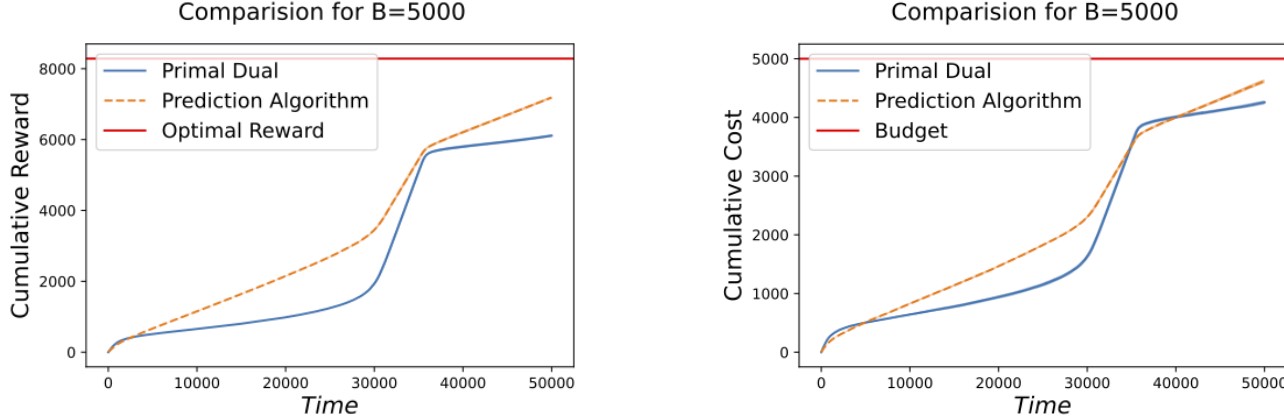

**Figure 6:** Experiments with $T = 50000, B = 5000, \pi = 0.5$. **Right:** full feedback rewards. **Left:** full feedback costs.

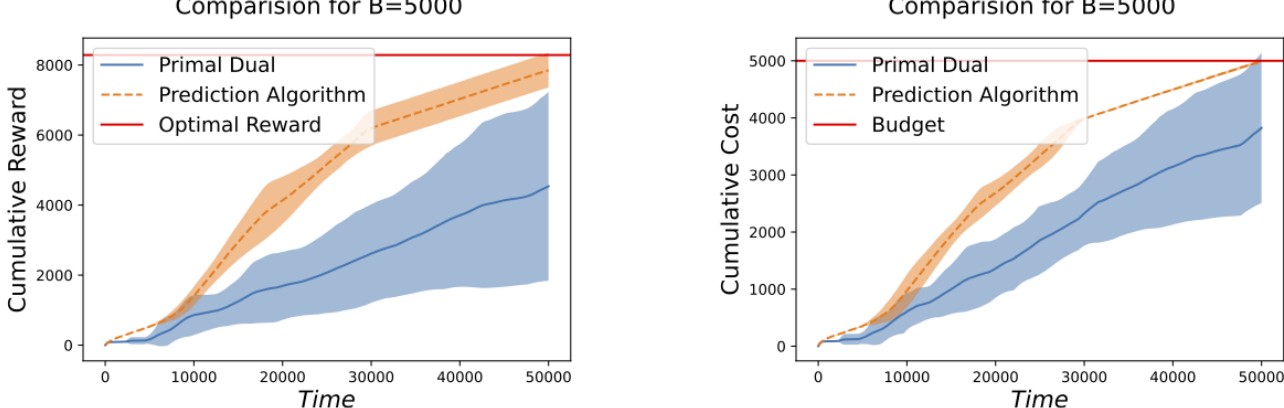

**Figure 7:** Experiments with $T = 50000, B = 5000, \pi = 0.5$. **Right:** bandit feedback, rewards. **Left:** bandit feedback, costs.

## A.2 REWARD ACROSS NOISE

In the following sections the noise is sampled from a distribution with $\mu = 0$ and $\sigma = x$ where $x$ is indicated on the $x$ axis. The noise is subsequently added to the probability vector, then the absolute value is taken and a normalization is performed. In Figures 8 and 9 we compare the performance of both Algorithm 2 and 3 in the same setting as the main text (i.e., with $T = 50000$, $B = 5000$ and $\pi = 0.9$) with the difference that we introduce a probability of skipping some iteration $\mu = 1/\sqrt{T}$, which will become useful in the adversarial setting. Such probability is not impacting heavily the performance of the algorithm, both in the full and bandit feedback settings.

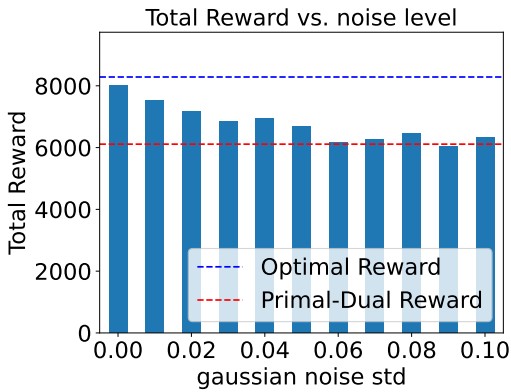

**Figure 8:** FF Across Noise with $T = 50000, B = 5000, \pi = 0.9, \mu = 1/\sqrt{T}$

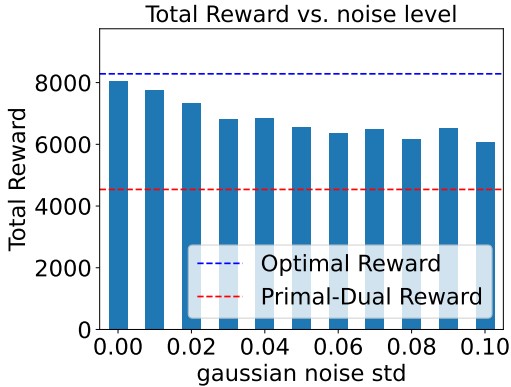

**Figure 9:** BANDIT Across Noise with $T = 50000, B = 5000, \pi = 0.9, \mu = 1/\sqrt{T}$

In Figures 10 and 11 we use a higher probability of skipping an iteration. The value used is $\mu = \frac{2\sqrt{2\log(1/\delta)}}{\rho T^{1/2}}$, using $\delta = 0.1$. Such value is theoretically correct, but becomes very high for lower values of $T$. Moreover, since we are testing on stochastic data, in practice such a high probability of skipping the iterations is not needed. Figure 10 shows how Algorithm 2 is still performing in a comparable way with respect to the primal-dual algorithm, even in a situation which is not suitable to showcase the importance of the theoretical value of $\mu$. Figure 11 shows how Algorithm 3 maintains a better performance than the primal-dual one. The underspending caused by not converging in the bandit setting, is worse than the underspending caused by the skipped iterations in Algorithm 3.

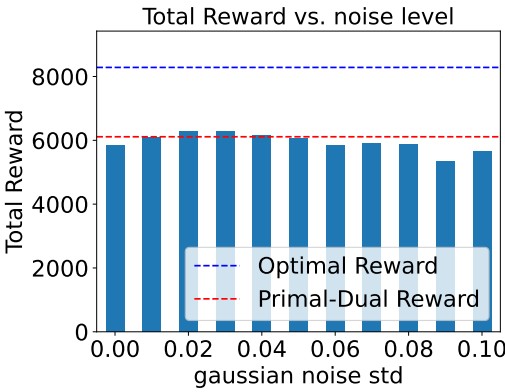

**Figure 10:** FF Across Noise with $T = 50000, B = 5000, \pi = 0.9, \mu = \frac{2\sqrt{2\log(1/\delta)}}{\rho T^{1/2}}, \delta = 0.1$

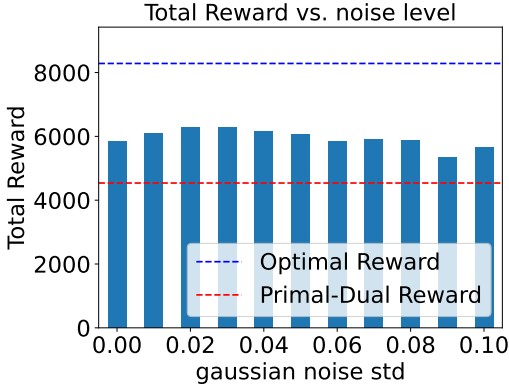

**Figure 11:** BANDIT Across Noise with $T = 50000, B = 5000, \pi = 0.9, \mu = \frac{2\sqrt{2\log(1/\delta)}}{\rho T^{1/2}}, \delta = 0.1$

## A.3 REWARD WITH HIGH NOISE

Figures 12 and 13 are important to show that increasing the amount of noise indefinitely, will not deteriorate the performance any further. Indeed, high noise causes the prediction to be an almost uniform probability vector, higher noise will not change the situation, on average. In particular, in Figure 13 it can be noticed how in such setting the bandit primal-dual algorithm performs worse than spending the budget on a random prediction. This is due to the data, but it is still an interesting observation.

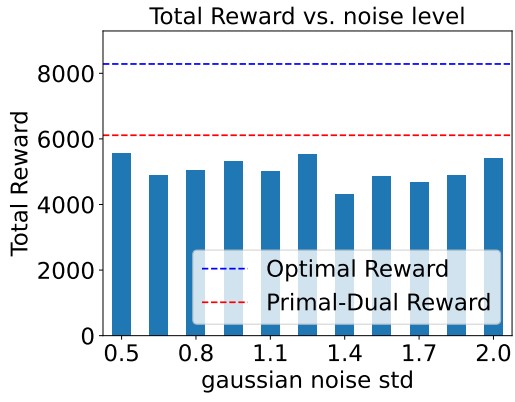

**Figure 12:** FF Across HIGH Noise with $T = 50000, B = 5000, \pi = 0.9$

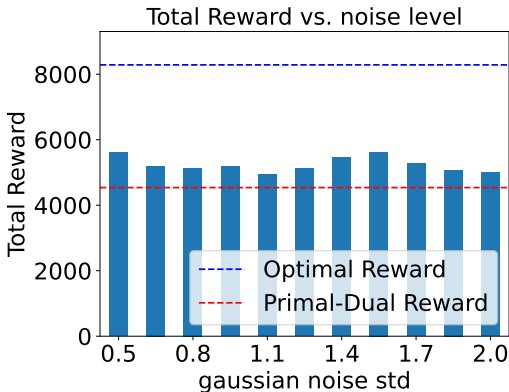

**Figure 13:** BANDIT Across HIGH Noise with $T = 50000, B = 5000, \pi = 0.9$

In the successive four graphs (Figures 14 to 17) instead, we see how the performance of the algorithms deteriorates when the noise is high and we also use the two values of $\mu$ described previously. The main thing to notice is that, as in the case for $\mu = 0$, the performance of the algorithms does not deteriorate any further with increasing noise.

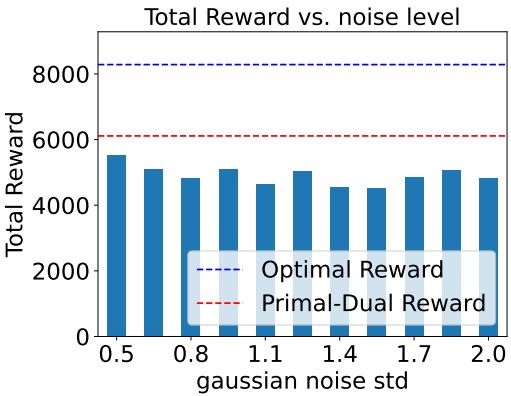

**Figure 14:** FF Across HIGH Noise with $T = 50000, B = 5000, \pi = 0.9, \mu = 1/\sqrt{T}$

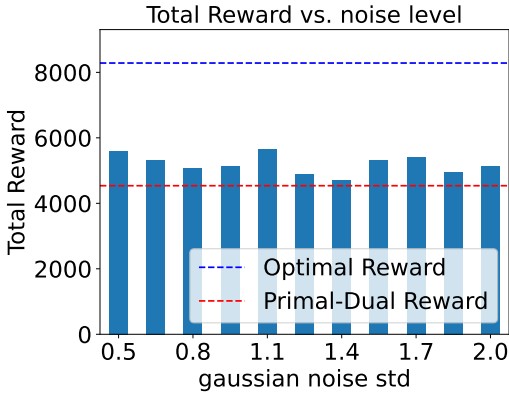

**Figure 15:** BANDIT Across HIGH Noise with $T = 50000, B = 5000, \pi = 0.9, \mu = 1/\sqrt{T}$

The last table shows the performance, in terms of rewards and costs, and the standard deviation of Algorithms 2 and 3, when the value of $\mu$ is set to the two fixed values. It is interesting to see how the high value of $\mu$ does not allow the budget to be fully spent, which in such case may result in a loss of reward, but in edge cases may result in the opposite outcome, therefore justifying its importance in the theoretical analysis.

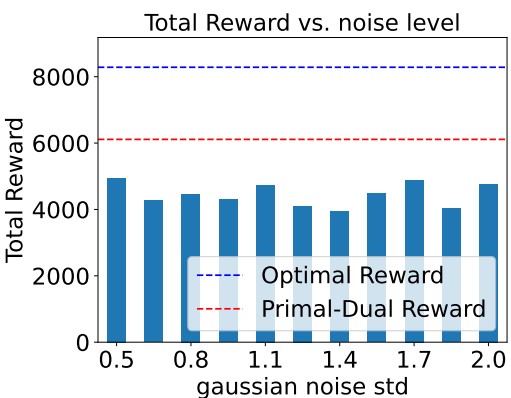

**Figure 16:** FF Across HIGH Noise with $T = 50000, B = 5000, \pi = 0.9, \mu = \frac{2\sqrt{2\log(1/\delta)}}{\rho T^{1/2}}, \delta = 0.1$

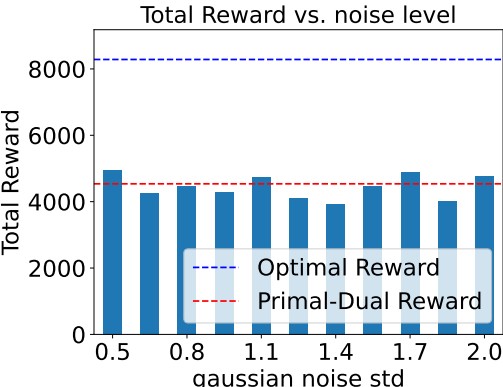

**Figure 17:** BANDIT Across HIGH Noise with $T = 50000, B = 5000, \pi = 0.9, \mu = \frac{2\sqrt{2\log(1/\delta)}}{\rho T^{1/2}}, \delta = 0.1$

| Algorithm | Reward | Cost | Std |
|---|---|---|---|
| Algorithm 2 (full feedback, $\mu_{low}$) | 8001.34 | 4862.58 | 18.87 |
| Algorithm 3 (bandit feedback, $\mu_{low}$) | 8060.11 | 4969.10 | 52.29 |
| Algorithm 2 (full feedback, $\mu_{high}$) | 5855.51 | 3526.10 | 19.81 |
| Algorithm 3 (bandit feedback, $\mu_{high}$) | 5855.51 | 3526.10 | 19.81 |

**Figure 18:** Performance with perfect prediction, with "$\mu_{low}$" meaning $\mu = 1/\sqrt{T}$ and "$\mu_{high}$" meaning $\frac{2\sqrt{2\log(1/\delta)}}{\rho T^{1/2}}, \delta = 0.1$