# OpenReview forum: "Bandits with Knapsacks and Predictions"
_auai.org/UAI/2024/Conference — UAI 2024 poster_

### Official Review · Reviewer_S4Qi · 2024-03-14

**Q2-1 Originality-Novelty:** 2
**Q2-2 Correctness-Technical Quality:** 3
**Q2-5 Clarity Of Writing:** 3

**Q1 Summary And Contributions:**

This paper studies bandits with knapsacks, and assume that the learner has some predictions of the optimal solution at the beginning. In this case, the authors design several algorithms. In the adversarial case, the algorithms follow the predicted solution with probability $\pi$, and do an online learning policy with probability $1-\pi$. In the stochastic case, the algorithm follows the predicted solution at the beginning, and changes to do an online learning policy only if the observations show that the predicted solution is not optimal. By this way, these algorithms achieve good performance when the predicted solution is exactly optimal, and not-too-bad performance when the predicted solution is not optimal. Finally, they use some experiments to demonstrate the effectiveness of their algorithms.

**Q2-3 Extent To Which Claims Are Supported By Evidence:**

3: Good: the main claims are supported by convincing evidence (in the form of adequate experimental evaluation, proofs, (pseudo-)code, references, assumptions).

**Q2-4 Reproducibility:**

3: Good: key resources (e.g. proofs, code, data) are available and key details (e.g. proofs, experimental setup) are sufficiently well-described for competent researchers to confidently reproduce the main results.

**Q3 Main Strengths:**

- The problem setting is well-motivated.
- The intuitive idea is simple but effective.
- There are some experiments demonstrating the algorithms' effectiveness.

**Q4 Main Weakness:**

- The results look incremental, and I do not see the main challenges in the analysis.
- Some parts require more clarification (see details below).

**Q5 Detailed Comments To The Authors:**

- In the stochastic case, the OPT is a per-step reward, but in the adversarial case, the OPT is a long-term reward? This seems not consistant.

- The authors claim that they show the tightness of their algorithms, however, I do not see the tightness after reading Appendix D. Please give more explanations about this.

- Could we use the gradient descent methods and treat the predicted solution as the start point?

- In line 7 of Algorithm 4, why function $h$ has two inputs?

- In the proof of observation 1, why the regret at the beginning is at most $h(s − 1)$?

**Q9 Complying With Reviewing Instructions:**

Yes

---

### Official Review · Reviewer_PQqg · 2024-03-23

**Q2-1 Originality-Novelty:** 3
**Q2-2 Correctness-Technical Quality:** 3
**Q2-5 Clarity Of Writing:** 3

**Q1 Summary And Contributions:**

The paper provides regret/competitive ratio gurantees for Bandits under Knapsack constraints for both stochastic/adversarial models when there is access to large offline data that can be used to enable future predictions of both rewards and the costs.

**Q2-3 Extent To Which Claims Are Supported By Evidence:**

3: Good: the main claims are supported by convincing evidence (in the form of adequate experimental evaluation, proofs, (pseudo-)code, references, assumptions).

**Q2-4 Reproducibility:**

1: Poor: key details (e.g. proof sketches, experimental setup) are incomplete/unclear, or key resources (e.g. proofs, code, data) are unavailable.

**Q3 Main Strengths:**

- The paper is well written - provides adequate motivation and highlights the technical challenges involved.
- To the best of my knowledge, the proof sketches in the main paper seem fine.
- The setup is practically motivated and as such improvement in regret/competitive ratio for the setting would be a worthful contribution to the community.

**Q4 Main Weakness:**

- The experiments are not reproducible as the code was not submitted.

**Q5 Detailed Comments To The Authors:**

1. The authors say "To prevent our algorithm from missing the items arriving in the last part of the input sequence, we sample the time steps served by ........  respectively". Could the authors explain the intuition behind the $1/sqrt{T}$ factor?

***

2. There are a few missing references to recent papers in Knapsacks:

    1) Reinfocement Learning with Knapsacks: https://arxiv.org/abs/2006.05051
    2) Smooth Adversarial Bandits with Knapsacks: https://proceedings.mlr.press/v162/sivakumar22a.html
    3) Dueling Bandits with Knapsacks: https://arxiv.org/abs/2312.17229
It might be a good idea to include these in the related works section.

***

3. Does the stochastic setting also assume $B = \Omega(T)$? Existing bounds (in the without prediction case need $B = \Omega(T^{3/4})$). Could the results in the paper be extended to a lower budget regime?

**Q9 Complying With Reviewing Instructions:**

Yes

---

### Official Review · Reviewer_BYJ7 · 2024-03-25

**Q2-1 Originality-Novelty:** 3
**Q2-2 Correctness-Technical Quality:** 3
**Q2-5 Clarity Of Writing:** 4

**Q10 Ethical Concerns:**

No.

**Q1 Summary And Contributions:**

The paper studies the Bandits with Knapsacks problem with the aim of designing a learning-augmented online learning algorithm upholding better regret guarantees than the state-of-the-art primal-dual algorithms with worst-case guarantees, under both stochastic and adversarial inputs.

**Q2-3 Extent To Which Claims Are Supported By Evidence:**

3: Good: the main claims are supported by convincing evidence (in the form of adequate experimental evaluation, proofs, (pseudo-)code, references, assumptions).

**Q2-4 Reproducibility:**

3: Good: key resources (e.g. proofs, code, data) are available and key details (e.g. proofs, experimental setup) are sufficiently well-described for competent researchers to confidently reproduce the main results.

**Q3 Main Strengths:**

The paper cleverly improved the existing state-of-the-art by incorporating a prediction algorithm. The paper is well written. The idea is novel. All theoretical and empirical results are clearly stated and justified.

**Q4 Main Weakness:**

See Q5.

**Q5 Detailed Comments To The Authors:**

1. How is $\ell_\tau$ defined in Algorithm 4? I did not find where this notation is introduced in the main text.
2. As the main contribution is considering a prediction algorithm on top of (Castiglioni et al., 2022b)'s, I suggest adding more comparison to that work. For example, adding emphasize on how the regret is improved given perfect prediction in adversarial setting.
3. What would be a practical motivating example of considering such prediction oracle? I think the paper does not have enough motivation of this setting.

**Q9 Complying With Reviewing Instructions:**

Yes

---

### Official Review · Reviewer_c7gC · 2024-03-25

**Q2-1 Originality-Novelty:** 2
**Q2-2 Correctness-Technical Quality:** 2
**Q2-5 Clarity Of Writing:** 2

**Q1 Summary And Contributions:**

This paper studied bandits with knapsacks. It analyzed the performance of the proposed Algorithm 2 in both stochastic and adversarial environments with either full feedback or bandit feedback. It also evaluated Algorithm 2 with numerical experiments.

**Q2-3 Extent To Which Claims Are Supported By Evidence:**

2: Fair: the main claims are somewhat supported by evidence (but the experimental evaluation may be weak, or does not match entirely with the claims, important baselines may be missing, proofs contain important ideas but lack rigor, algorithmic details are only discussed superficially, references are imprecise, assumptions are not sufficiently motivated or explicated, etc.).

**Q2-4 Reproducibility:**

3: Good: key resources (e.g. proofs, code, data) are available and key details (e.g. proofs, experimental setup) are sufficiently well-described for competent researchers to confidently reproduce the main results.

**Q3 Main Strengths:**

It analyzed the performance of the proposed Algorithm 2 in both stochastic and adversarial environments with either full feedback or bandit feedback. It also evaluated Algorithm 2 with numerical experiments.

**Q4 Main Weakness:**

1. The terms 'competitive ratio' and 'best fixed strategy prediction', and the parameter $\pi$ and $\rho$ are key elements of the results. May the authors explain them before or around Theorem 1.1? The current explanation is a bit far away from the first time they are mentioned.
    1. For instance, the definition of '$1/c$-competitive' is in Section 3. Is $1/c$ is the so-called 'competitive ratio'?
1. Based on the existing works, what is the literature gap this paper attempted to fill?
1. As the full-feedback setting is studied in this work, may the author(s) discuss related works? I don't see any in the paper.
1. The results from existing literature should be compared in a table.
1. Section 2: the difference between stochastic and adversarial environments does not seem to be clear.
1. Algorithm 1: how does $R^P$ and $R^D$ be updated? What is the output?
1. What do $ ALG^{WC}.next\underline{\hspace{0.5em}}  element()$ and $ ALG^{WC}.observe\underline{\hspace{0.5em}} utility()$ mean?
1. Section 2: It stated that 'The case of $m > 1$ can be addressed with minor modifications'. I suggest the author(s) to provide evidence and results regarding this statement.
1. What does the 'perfect prediction' in Lemma 3.2 indicate?
1. There should be more discussions on the theoretical results. Compare to existing algorithms/lower bound?
1. The experiment does not compared the proposed algorithm to existing algorithms. The superiority of this work is not clear.

**Q5 Detailed Comments To The Authors:**

Except for the major concerns listed in Q4, some minor issues are as below:
1. The citation type in beginning of Section 1.2: Should be ' ... Chapter 10 of Slivkins et al. [2019] ...'
2. The author(s) may consider to revise some long sentences, such as:
    1. Section 1.2: 'The initial guarantees of $\alpha = O(m\log T)$ by Immorlica et al. [2019] where later improved to a tight $O(\log m\log T)$ by Kesselheim and Singla [2020] and, in the case of $B = \Omega(T)$, Castiglioni et al. [2022b] showed how to achieve a constant competitive ratio of $1/\rho$.'
    1. Section 1.2: 'Applications of this framework are wide-ranging and include scheduling, as highlighted by significant contributions in the literature by Mitzenmacher [2019], Lattanzi et al. [2020], and caching or paging algorithms, as explored in works such as [Rohatgi, 2019].'
    1. Section 2: 'When rewards and costs are stochastic the baseline is the best fixed randomized strategy that satisfies the constraints in expectation, which is the standard choice in stochastic BwK settings [Badanidiyuru et al., 2018, Castiglioni et al., 2022b]: ... where expectations are with respect to randomness of the environment when generating rewards and costs.'

**Q9 Complying With Reviewing Instructions:**

Yes

---

### Official Review · Reviewer_PaAF · 2024-03-26

**Q2-1 Originality-Novelty:** 2
**Q2-2 Correctness-Technical Quality:** 2
**Q2-5 Clarity Of Writing:** 2

**Q1 Summary And Contributions:**

The authors delve into the Bandits with Knapsacks problem, aiming to create a learning-augmented online learning algorithm that outperforms existing primal-dual algorithms with worst-case guarantees. Their focus extends to both stochastic and adversarial inputs.

**Q2-3 Extent To Which Claims Are Supported By Evidence:**

2: Fair: the main claims are somewhat supported by evidence (but the experimental evaluation may be weak, or does not match entirely with the claims, important baselines may be missing, proofs contain important ideas but lack rigor, algorithmic details are only discussed superficially, references are imprecise, assumptions are not sufficiently motivated or explicated, etc.).

**Q2-4 Reproducibility:**

2: Fair: key resources (e.g. proofs, code, data) are unavailable but key details (e.g. proof sketches, experimental setup) are sufficiently well-described for an expert to confidently reproduce the main results.

**Q3 Main Strengths:**

- The problem is interesting and relevant within the bandits field.
- The theoretical results are noteworthy.
- The assumptions are fairly mild.
- The readability is good.

**Q4 Main Weakness:**

- The theoretical analysis could be made more thorough and clear.
- The experiments using small test examples are helpful. However, it would make the experiments richer if some real-world application(s) as pointed out in the introduction of the paper were used for the experiments.
- Runtime and space analysis of the algorithms should be done.

**Q5 Detailed Comments To The Authors:**

Please refer to the weaknesses above. Furthermore,

- You use the terminology *full* and *bandit* feedback settings. What do you mean by these terms, namely *full* and *bandit*? I believe by *full* you mean *semi-bandit* feedback and by *bandit* you mean *full-bandit* feedback. A clarification would be helpful.
- In my opinion, Our Results section can be deferred after related work and a little bit of setting up the problem. As of now, it's too technical to be there quite at the beginning of the paper.
- What do you mean by the smoothness of an algorithm? Is it related to robustness?

**Q9 Complying With Reviewing Instructions:**

Yes

---

### Meta-Review · Area_Chair_kcYL · 2024-04-14

This paper investigates the Bandits with Knapsacks problem, aiming to develop a learning-augmented online learning algorithm. The authors propose several new algorithms and offer theoretical guarantees in the form of competitive ratios and regret bounds. However, reviewers have expressed concerns regarding the novelty and several technical details. Given the high competitiveness of this year's UAI conference, the authors may have to polish and improve the paper by carefully considering the reviewers' feedback.